# Multi-Criterial Algorithm for the Efficient and Ergonomic Manual Assembly Process

**DOI:** 10.3390/ijerph19063496

**Published:** 2022-03-16

**Authors:** Maja Turk, Marko Šimic, Miha Pipan, Niko Herakovič

**Affiliations:** Faculty of Mechanical Engineering, University of Ljubljana, Aškerčeva Cesta 6, 1000 Ljubljana, Slovenia; maja.turk@fs.uni-lj.si (M.T.); miha.pipan@fs.uni-lj.si (M.P.); niko.herakovic@fs.uni-lj.si (N.H.)

**Keywords:** digitalization, Industry 4.0, multi-criterial algorithm, smart technologies and tools, ergonomic workplace, productivity, assistance system

## Abstract

Industry 4.0 introduces smart solutions throughout the company’s supply chain, including manual assembly, where the goal is to ensure shorter work cycle time, increase productivity and quality, while minimizing costs. Following the principles of this paradigm, this paper proposes a digital transformation of the manual assembly process by implementing a multi-criterial algorithm (MCA) for adjusting and configuring a human-centered smart manual assembly workstation to ensure efficient and ergonomic performance of the manual assembly process. The MCA takes into account various influential parameters, such as the anthropometry of the individual worker, gender, complexity of the assembly process, product characteristics, and product structure. The efficiency of the MCA was verified both in the laboratory environment with the time analysis and in the virtual environment using Digital Human Modelling through several ergonomic analyses. The results of the implementation of the MCA on a manual assembly workstation support the digital (re)design of the manual assembly process with the aim of creating an efficient and ergonomically suitable workstation for each worker, thus increasing the productivity and efficiency of the human-centered manual assembly process.

## 1. Introduction

In the 21st century, manufacturing companies face unpredictable and unforeseen changes in the market dictated by global competition. These changes include the rapid introduction of new products and the constant change in their quantity [1,2]. In recent years, a new trend has emerged both in industry and in other sectors, which is called the fourth industrial revolution or Industry 4.0 (I4.0) [3,4]. The I4.0 paradigm has introduced a new term “smart”, which is considered a key feature of the production systems of the future [5].

Manufacturing is one of the most important industrial processes, where the initial material is transformed into the final product through the production process, which includes work equipment (machines), tools, and people (workers). One of the manufacturing processes, relevant for the study is the (manual) assembly process, which involves the composing of previously manufactured components and/or sub-assemblies into a complete product of unit of a product, primarily performed by human operators using their inherent dexterity, skill, and judgement [6]. In light of I4.0, changes are particularly taking place in the field of (manual) assembly, focusing on reducing the workload of workers [5] through the digital transformation of the workplace [7,8,9]. Digital workplace design aims to facilitate employees’ current and future work practices through digital technologies [9]. Proper digital workplace design is critical to sustainable business success in a new digital, consumer-centric business world. The digital workplace impacts physical workplaces, technology, and people and it is widely recognized to optimize worker productivity [7]. The transformation of the digital workplace goes beyond the adoption or non-adoption of technology—it has far more profound implications related to workplace redesign. Future digital work not only means changing the tools used in work activities, but it often changes the nature of work activities and processes themselves [8]. As we know, many manufacturing jobs still require manual labor involving a variety of activities, such as assembly, loading and unloading, pushing and pulling, and performing tasks that require manual material handling [10], and these are the activities and jobs that need to be digitally transformed.

Competencies, skills, abilities, and satisfaction of workers are very important conditions for increasing the productivity of production [11,12,13]. Therefore, it is necessary to develop highly skilled workers capable of performing multiple tasks [14], or a human-centered manual assembly system with digital instructions [15,16] and/or a virtual worker training system [17] that replaces extensive training and the lack of experience of the highly skilled workers.

On-the-job training and support are essential to help workers acquire the necessary skills and competencies, improve worker productivity, and ensure product quality [18]. In addition to competencies and skills, worker performance is closely linked to their work environment. Product and workplace design should take into account the information gathered about processes, tools, machines, work items, tasks, and workers, often taking into account conflicting constraints and designing workplaces that are acceptable to all stakeholders [19]. Redesigning existing workstations or designing new modern manual assembly workstations according to ergonomic specifications allows workers to work safely, reducing the risk of work-related musculoskeletal disorders and avoiding potentially hazardous work movements and stresses [20]. In recent years, computer-based techniques for ergonomic assessment of work tasks have been widely used, especially virtual human simulation (DHM, Digital Human Modelling). DHM tools allow rapid virtual prototyping and evaluation of the considered workstation configuration using the technique of “what-if” scenarios without exposing the worker to risk [21] and without investing in physical materials and resources [22]. In this way, DHM lends strong scientific validity to proposed and already implemented improvements and solutions in manual assembly and workplace design [6].

“Smart” assembly means that the assembly system must be adapted using smart actuators as a result of generated information and in accordance with algorithms [23,24]. Based on the definition of a “smart system”, Bortolini et al. [25] propose a general framework that presents the implications of I4.0 principles on the development and design of an assembly system called “Assembly System 4.0”. The main features of the proposed systems are customization to meet the individual requirements of customers of products at a later stage of development, an assembly control system, computer-aided assembly, intelligent inventory management, traceability of products and processes, and self-configuration of a manual assembly workstation. Cohen et al. [24] have presented a general architectural model for implementing I4.0 principles in an existing assembly system, focusing on the Operator Support System (OSS) and the Self-Adapting Smart Assembly System (SASS). OSS refers to real-time data and information that are constantly available to the operator, and SASS refers to the adaptation of the work process by actuators, as a result of the obtained data and smart algorithms. The authors emphasize that the implementation of OSS and SASS has an impact on increasing flexibility, agility, scalability, and productivity of a smart assembly system.

I4.0 principles, ergonomic design, and digital transformation of the work environment can be most easily ensured by implementing smart technologies and tools at (manual assembly) workstations. Papetti et al. [26] propose a redesign of the manual workplace according to ergonomic recommendations in the footwear industry. They emphasize redesigning the work environment from a task-based organizational model to a human-centered model that promotes the development of personal skills and worker well-being. The paper presents the redesign of the hand sewing workstation after ergonomic evaluation by experts. The experts were asked to perform a heuristic evaluation based on direct and video-based observation of users according to standardized ergonomics assessment methods (e.g., RULA, OCRA, and NIOSH). Workers were interviewed to take into account their preferences and satisfaction with the workplace to better interpret the analysis results. Biometric and environmental parameters are automatically collected through the use of appropriate digital tools and IoT devices. As a result of the study, workplaces were effectively redesigned, mitigating the highest ergonomic risks without compromising the performance of value-added work. Weyer et al. [3] presented the concept of a smart factory with a computerized manual assembly workstation designed to assemble small components using augmented reality (AR) and advanced sensor technology. The concept follows the paradigm of the “enriched operator” and enables flexible and modular integration into automated production lines. Using RFID (radio frequency identification) technology, the appropriate information and assembly instructions are obtained from the input product or raw material, enabling mass production of highly customized products. Workers are assisted by virtual instructions (AR) directly at the assembly workstation using tablets or smart glasses, while work progress at the manual assembly station is monitored by a static 3D camera. Thus, they enabled synchronized work of manual and automated assembly according to the concept of “man in a loop”. Bertram et al. [14] give an overview of existing solutions and prototypes in the field of assistance systems for manual workstations in research and practice and discuss their specific focus. All systems focus on an information assistant that provides the right information for the right situation. Only a few solutions and projects are presented here in the article: Augmented Workplace from the motionEAP project (development of an augmented workstation), ProMiMo (implementation of a user-centered assistance for manual assembly at a workbench), Manual Working Station from SmartFactory KL (development of a smart workstation, equipped with an assistance system for the sequential assembly process), Operator Support System TNO (development of an operator support system for assembly that focuses on information assistance for the worker), and Active Assist Bosch Rexroth (system that serves as a configurable and open web platform. Features of this software include contextual information provision and a standardized interface for additional system components (e.g., pick-to-light, projector, touch screen, RFID reader)). Zamfirescu et al. [27] focused on visualization and digital guidance of the manual assembly process. They claim that with increasing complexity and variability of products, the main difficulty for the operator is to follow the correct assembly procedure to correctly produce the desired product. To assist in the process, there are many possible alternatives for establishing a smart work environment (e.g., pick/put-by-light/voice/vision systems) to guide the human operator during the assembly process. There are also alternative recommendations, such as augmented reality (AR), virtual reality (VR) guidance [15,18], and projector-based digital guidance [28]. The evaluation of AR visualization assistive systems in the reported papers consistently found that untrained users can assemble products faster and with a lower error rate, which is the goal in industry and digital transformation theory. In the literature, less attention has been paid to the microlevel of digital transformation, i.e., the individual workplace environment and the range of new digital tools that support or hinder the way people work [29].

From the literature [5,11,12,13,18,19], it is clear that the need for human labor is great and that only with a properly designed work environment is it possible to perform tasks efficiently and ergonomically, thus increasing productivity in industry. In order to achieve an ergonomic manual assembly process and remain competitive in the market, it is necessary to develop a new algorithm that controls self-configuring and self-adaptive smart manual assembly workstations implemented with smart tools and technologies. The idea of self-configuration comes from the principles of I 4.0, which is the restructuring of today’s systems into smart factories.

The market offers us many smart solutions and tools, which in themselves do not guarantee that the manual assembly workstation will be self-configuring and adaptable. We also do not find complete solutions on the market that would provide an ergonomically designed smart manual assembly workstation for an individual worker. When designing a smart manual assembly workstation, various parameters must be taken into account, such as: type of product (dimensions, material), type of assembly (high, medium, low complexity), product structure (component assembly sequence), gender of workers, anthropometric data of workers (age, body height, limitations), type of workplace (standing, sitting, combined), etc. The solutions of individual providers in the market are partial, as they only offer solutions for specific segments described above [30,31]. Thus, the industry still has the problem of an inadequately designed workplace that cannot keep up with rapid market changes, “customization” of products, and the constant tendency to (re)configure the workplace in relation to the individual worker.

One of the possible solutions to overcome this problem and the main contribution of our study is the digital transformation of the workplace through the implementation of a new multi-criterial algorithm (MCA) on the manual assembly workstation. Digital transformation has been found to come in many shapes and forms. It is therefore not surprising that strategies for digital transformation take a broad perspective and emphasize the transformation of products, processes, and organizational aspects due to new technologies. In our paper, we propose the digital transformation of the manual assembly process by digitalization of the manual assembly workstation through the development and implementation of a new algorithm. An MCA configures the workstation and guides the worker according to the assembly instructions. It takes into account several influential parameters that need to be considered when (re)designing a new assembly workstation to make it smart, self-configuring, and ergonomic. Influential parameters that are considered are: the nature and skills of the individual workers, the characteristics of the products, and the complexity of the assembly. Only through the “right” combination of parameters, provided by an MCA that controls smart tools and adapts the workplace to the individual worker, we can achieve a significant increase in productivity and an ergonomically suitable workplace. At the same time, an MCA facilitates the prevention of errors during the assembly process (even for untrained workers), thus increasing the value added per worker.

This paper is organized as follows: Section 2 focuses on the step-by-step presentation of the MCA and the description of the case study. Section 3 presents the results and discussions on the error analysis and ergonomic evaluation of the workstations. Section 4 summarizes the final results of the study.

## 2. Materials and Methods

This section describes the development and operation of the MCA. The MCA is written in the open-source Python version 3.7 programming language and controls the entire smart manual assembly workstation with implemented smart technologies and tools. The algorithm guides the individual worker through the entire assembly process with digital instructions and corresponding visualization. As stated in Qu et al. [32], one of the main functions of the smart manufacturing systems is the self-adaptive function. The self-adaptive function controls the behavior of the elements of smart manufacturing systems based on real-time sensor data and information. Through the various algorithms and rules, the self-adaptive behavior of smart manufacturing systems is regarded as continuous learning or lifelong learning. As the primary intelligence level, adaptivity implies the ability to act according to rules. “If-Then-Else” is the primary rule we follow in developing our algorithm.

Section 2 is divided into two parts. The first part contains the description and development of the MCA and the second part presents the overview of the case study, which includes a description of the experimental environment and the characteristics of the products and workers who performed the experiment.

### 2.1. Multi-Criterial Algorithm

According to the literature review, manual assembly is mainly evolving towards AR [15,18,28], but visualization itself through AR does not ensure higher productivity and ergonomic suitability of the workplace. To achieve these results, it is necessary to develop a self-configurable smart manual assembly workstation with smart technologies for error prevention, visualization, and an implemented MCA that combines the technologies and adapts the workstation depending on the selected influence parameters. The new algorithm includes three main groups of influence parameters: human, assembly process, and product, which are stored in a SQL Express database.

Figure 1 shows the block diagram of the MCA and the process of establishing communication with the hardware and RFID reader, the process of reading and writing data to the SQL Express database that controls the lighting, grab containers, pick-by-light laser, and rotational assembly nest, and the height of the worktable of the smart manual assembly workstation, the process of calculating the hardware parameters, and the process of adjusting the hardware of the workstation according to the calculation.

The main function of the proposed MCA is to adjust and configure smart assembly workstation and to control tools according to worker, assembly complexity, and product characteristics. The algorithm first establishes communication with the hardware (Raspberry Pi (RPi), sensors, RFID card). This is followed by a “while true” loop that ensures that the smart manual assembly workstation configures itself according to the desired parameters (coordinates or labels of the next parameters: worker’s ID, gender, body height, complexity of the assembly process, dimensions of the part, dimensions along the *Z*-axis (height), and product structure). From the SQL Express database, the new algorithm reads the current parameters (coordinates/labels) to control the following components: lighting, grab containers, laser indicator and rotational assembly nest, and worktable height. The algorithm calculates the new desired parameters (coordinates) of the controlled components based on the influence parameters. Which influence parameters are taken into account and the relationship between the parameters and the components of the manual assembly workstation is shown separately for each smart tool and technology on the right side of Figure 1. After calculating the parameters, the algorithm sends commands about the new coordinates of the components to the hardware, which responds to the commands by reconfiguring itself into the desired configuration. After 2 s of waiting, the “while true” loop is repeated. The smart manual assembly workstation configures itself the next time the program reads the new parameter value from the SQL Express database (the next step in the product assembly sequence).

The MCA took into account the influential parameters of the worker, the type of assembly process, and the product [33]. The worker’s gender and anthropometric characteristics, described only as a function of body height, were considered. The type of assembly process was divided into three complexities: normal assembly (assembly line work, assembly of simple products), precise assembly (assembly of smaller parts), and non-demanding (heavy) assembly (assembly of heavier and larger components, woodworking). Influential product parameters were described with dimensions of the whole product, dimensions of the product along the *Z*-axis, structure of the product (assembly sequence), and assembly direction.

Influential parameters have different effects on the configuration of the manual assembly workstation, or in other words, they have different effects on the adaptation of the smart tools, as shown on the right side of Figure 1.

For lighting adaptation, we focus on the intensity and direction of lighting. The intensity of the lighting is affected by the complexity of the assembly process. So, if the complexity is more demanding, we need higher light intensity, if the complexity is not demanding, the light intensity can be weaker. The lighting direction is influenced by the dimensions of the product along the *Z*-axis, because we want to have an illuminated product position at all times where we are assembling the current part. If the current part is assembled at a height of, for example, 100 mm from the starting point along the *Z*-axis, the lighting direction must be adjusted accordingly.

The grab containers are adjusted in distance and tilt to the height and gender of the worker. If the worker is smaller, the grab containers will move closer to him than if the worker is taller. Additionally, the grab containers will move closer if the worker is female, as women generally have a shorter arm span. The tilt of the grab containers is adjusted to the worker so that they are tilted at a 20° angle to the table surface. However, the laser indicator depends on the structure of the product. Its task is to show the worker (mark with a laser beam) the part in the grab container that is being assembled in the current step of the assembly process.

Similar to a laser pointer, a rotational assembly nest depends on the structure of the product. According to the structure of the product and the optimal orientation of the part being assembled, the rotational nest rotates in a way that most easily allows the worker to complete the assembly operation.

The tool, which is adjusted according to the influential parameters, is the height of the worktable, which depends primarily on the worker. The relationship between the height and the elbow height of an individual worker is determined taking into account the height and gender. To determine the working height (calculation of working height 1 in Figure 1), 50 to 100 mm are subtracted from the elbow height according to the standard deviation, since the working height must be lower than the elbow height. Then, the height of the table increases or decreases depending on the complexity of the assembly process. If the complexity of the assembly is high, the height of the worktable must be higher, otherwise the working height will be lower. Calculation of working height 1.1 from Figure 1 takes into account the complexity of assembly process and represents the height of the worktable at the beginning of the assembly process. Taking into account the assembly of the product at height, the worktable height is reduced during the assembly process according to the dimensions of the product along the *Z*-axis, resulting in a working height of »1.2« from Figure 1. The determination of the influential parameters and their dependence on the configuration of the manual assembly workstation are described in more detail in the paper [33].

### 2.2. Overview of the Case Study

The research aims to present the functionality of an MCA that controls the manual assembly workstation with smart tools and technologies, and to obtain results about the efficiency and ergonomics of the manual assembly process performed with the support of the new proposed algorithm (Figure 2).

#### 2.2.1. The Course of the Experiment

For the experiment, the assembly process of eight different products was performed at a classic and a smart manual assembly workstation in a laboratory environment. The classic and smart manual assembly workstations are located in separate rooms so that the subjects do not interfere with each other during the experiment. The subjects performed the assembly processes of individual products one after another, e.g., assembly of products P1–P4 at a classic manual assembly workstation and then assembly of products P5–P8 at a smart manual assembly workstation. Before and after the experiment, the subjects had no contact with each other, so no information about the parts and the assembly structure was transferred in order to achieve the most realistic results. The experiments were recorded for further processing.

#### 2.2.2. Functionality of a Classic and Smart Manual Assembly Workstation

The laboratory experiment was conducted at a classic and a smart assembly workstation. Each workstation has different functionalities and implemented technologies and tools.

The functional features of a classic manual assembly workstation are: (i) a seated workstation, (ii) a height-adjustable chair, (iii) a fixed worktable height, (iv) a fixed storage position and tilt, (v) industrial lighting, (vi) paper instructions that the worker must turn over according to the sequence of assembly operations, and (vii) a fixed assembly nest.

The functional features and implemented tools of the smart assembly workstation, in addition to the MCA, are: (i) optional sit/stand workstation, (ii) height-adjustable worktable, and (iii) distance- and tilt-adjustable grab containers. The grab container moves into the reach of the worker if it contains a part that is in the current step of the product structure or moves out of reach if it does not contain the required part. To avoid errors in locating parts in the grab containers, the manual assembly workstation also features (iv) pick-by-light technology (laser indication). The laser beam guides the worker through the structure of the product by pointing to the area of the grab container where the current part is located.

Other technologies and tools incorporated into the smart manual assembly workstation include: (v) adaptive lighting depending on the intensity and direction of the light beam, (vi) a rotational assembly nest that rotates according to the optimal assembly direction, and (vii) interactive digital instructions, which guide the worker through the structure of the product (assembly process). In addition to assembly, the worker’s task is to press the “Next” button after each successfully completed step of the assembly process. AR/VR technology and a collaborative robot can also be integrated as separate units in the smart manual assembly workstation.

#### 2.2.3. Characteristics of Products and Workers

Eight different products were selected for the experiment. Table 1 describes the properties of the products P1 to P8. All products are assembled from LEGO blocks and configured as shown in the second column of Table 1. Table 1 also contains information about the number of parts of each product and the number of different parts needed to assemble the product. It also includes the dimensions and height (*Z*-axis) of the products. We have selected products that have different levels of complexity (different numbers of parts) and variability, but are still representative and easy to assemble. In this case, we define the complexity of the products as: (1) Precise (P) for higher number of small parts (more than 200 and size bellow 2 mm that require special assembly accessories, precise and demanding assembly; (2) Normal (N) for number of parts between up to 200 and parts size more than 2 mm; and (3) Heavy (H) for the large parts and especially high-mass parts. The assembly of the products was performed by untrained workers (regardless of gender, age, or anthropometric characteristics). At each assembly workstation (classic and smart), 160 experiments were conducted, giving us a total of 320 assembly process results. The experiments were conducted by PhD students and laboratory staff. They conducted the experiment for the first time and were recruited voluntarily. The experiment was performed by 40 subjects (N = 40, gender: 26 male, 14 female). Their mean (SD) anthropometric data were: Age 31.7 (7.4) years; Height 1780 (90 mm). Each was instructed to perform the experiment at a rate that was feasible for the entire work shift, i.e., a person’s entire working life.

#### 2.2.4. Evaluation Process

To evaluate the efficiency and productivity of the manual assembly process, we recorded each subject during the assembly process for each product. Then, we performed a time analysis, determining the average time with standard deviation for each assembly process. The average times obtained at a classic and a smart manual assembly workstation for the same product were then compared.

To evaluate ergonomics, we used the Siemens Jack DHM software package, which is in our opinion one of the important approaches and strong supporting technologies closely related to I4.0, the modelling, simulation, and evaluation of processes in the virtual environment. For this purpose, we imported computer-aided design (CAD) models of both workstations. We used a male avatar with a height of 1780 mm and a female avatar with a height of 1770 mm as virtual humans and performed an ergonomic evaluation for the entire assembly process of the P1 product. In the virtual environment, we performed a reach envelope analysis, an analysis of the distribution of parts by grab containers according to areas (A—near area; B—repetitive range area; C—intermittent range area), and an analysis of the joint strain during the assembly process when assembling the product with or without MCA and tools (basic version vs. improved version).

## 3. Results and Discussion

The results and discussion section is divided into two main subsections: results of the multi-criterial algorithm (MCA) and results of the experiment based on time analysis and ergonomic evaluation. The MCA, the reconfiguration capabilities of the manual assembly workstation and the virtual approach to analyze the ergonomics are treated here as the main key-enabling technologies closely related to the I4.0.

### 3.1. Multi-Criterial Algorithm

The first key-enabling technology of I4.0 to transform the conventional manual assembly workstation into a smart manual assembly station represents the multi-criterial algorithm (MCA) that enables the proper reconfiguration of the assembly place including the individual components such as worktable height, lighting, assembly nest orientation, grab container setting according to the reachable criteria, and laser pointing the parts that cover the pick-by-light technology.

Figure 3 shows the block diagram of an MCA for step 2 of the assembly process for the actual worker who participated in the research. The values determined by the MCA for the individual configuration of the controlled components of the smart assembly workstation are calculated using empirically determined equations, described in [33]. The parameters read from the database are divided into three groups—worker, product, and complexity. As worker’s data, we use the ID of the worker: 1001, named Maja Turk, her height (1700 mm) and gender. Product data from the database are the “name” or type of LEGO block, its dimension, the dimension along the *Z*-axis, and step of the assembly operation. For complexity, we use the label N, which stands for the normal assembly process. The results of the new algorithm show the configuration of the smart manual assembly workstation, which is shown on the right side of Figure 3: (1) grab container: left container, distance: 500 mm from the shoulder, angle: 20°, section: 1; (2) height of the worktable: 905 mm; (3) lighting: intensity: 830 lux, direction: center of the assembly nest; and (4) rotation of the assembly nest: angle: 90°.

### 3.2. Laboratory Case Study Experiment

This section is divided into two parts: time analysis of the assembly process conducted at classic and smart assembly workstations and ergonomic evaluation of manual assembly process.

#### 3.2.1. Time Analysis

The results of the comparison of assembly times for the assembly of products from P1 to P8 on classic and smart manual workstations are shown in Table 2. At the smart manual workstation, all workers completed assembly faster than at the classic workstation. For example, product P1 is assembled in 216.8 s on average at the classic manual workstation, which is 14.4% longer than at the smart workstation (185.3 s). Product P8 takes workers at the classic workstation an average of 1082.3 s, which is 11.3% longer than at the smart workstation (959.8 s). If we generalize the results for all the products in the experiment, we find that the time saved for each product ranges from 11.3% to 14.9%, or on average 13.6% of the time for each product assembled at a smart manual workstation. At this point, it should be emphasized that the time savings are only the result of the implementation of the new I4.0-related technologies and were not the primary goal of the research. Thus, we do not want workers to have to follow stricter time norms. Therefore, we can say that by introducing an MCA and implementing smart tools at the manual assembly workstation, we increased the productivity and efficiency of workers, regardless of their prior knowledge of the manual assembly process or experience.

#### 3.2.2. Ergonomic Evaluation

The ergonomic evaluation as one of the important approaches of I4.0 was performed by three different analyses: (i) reach envelope analysis, (ii) distribution analysis of parts in grab containers (forward reaching), and (iii) joint strain analysis during the assembly process. All ergonomic analyses were performed in the Siemens Jack DHM environment. We imported CAD models of manual assembly workstations into the virtual environment, added a predefined avatar, and designed the entire assembly process as in a real environment.

The first ergonomic analysis we performed was determination of reach envelope. With the reach envelope, it is possible to verify the ergonomic and design suitability of the assembly process and the smart manual assembly workstation for any combination of worker and configuration of the smart manual assembly workstation. The reach analysis creates and displays the area (envelope) of maximum reach that is comfortable for an individual worker to perform activities, more specifically, a range that does not pose a risk of work-related illness and injury. In Figure 4, we see that all sections of the grab containers for both workers (male and female) are within the reach envelope, which means that the position of the grab containers, calculated by the MCA, is ergonomically suitable for long-term work without risk of illness and injury.

The second ergonomic analysis represents the distribution of forward reaches per work cycle. The analysis of forward reaches was performed for the assembly process of entire product P1 with 34 parts. For the case of the smart manual assembly workstation, we already considered ergonomics using an MCA and empirical equations based on the worker’s gender and height. Figure 5 shows the number of forward reaches into each ergonomic area (A, B, C) for a male (Figure 5, left) and a female (Figure 5, right) avatar at the smart manual assembly workstation. The worker never enters the danger zone (far reach, area C), so the position of the grab container complies with the ergonomic recommendations. We can claim that by adjusting the grab containers according to the MCA, taking into account the reach of the hands, gender, body height, and the position of the part in the grab container, we eliminate the forward reaches into the danger zone (area C). Each worker mainly reaches forward into the acceptable zone (area B), which does not affect the deterioration of the worker’s health in the long run. Comparing a male and a female avatar, a male reaches less often into area B because of a longer hand range than a female. Only a male avatar can be used for graphical presentation of the results for both the male and female.

The third ergonomic analysis we performed is the analysis of the load on the joints (joint strain) during the assembly of the product P1. We compared two versions of the process of manual assembly in a simulation environment, the first without the MCA and smart tools (basic version) and the second with them (improved version).

The results of the time during which the joints are strained (Table 3) are shown in pie charts, where orange represents an overloaded joint, yellow represents a conditionally acceptable loaded joint, and green represents permissibly loaded joint. Table 3 only shows some selected results. The goal for improving the basic version is primarily to eliminate or reduce the percentage of time in which joints are unacceptably loaded (orange), although this may consequently increase the percentage of time in conditionally acceptable (yellow) load for certain joints. When adjusting the load of the joints according to the ergonomic recommendations, in the improved version we focused on avoiding the rotation of the joints and unnatural deviations that are present in the basic version. For all joints, we can see a significant improvement in the percentage of time that the joint is subjected to unacceptable loads. Therefore, we have successfully reduced or eliminated the unacceptable movements during the manual assembly process in terms of health risk.

## 4. Conclusions

Nowadays, there is a need to restructure companies according to the principles of Industry 4.0 in order to ensure higher productivity and manufacturing efficiency, while introducing ergonomic working conditions. Despite the modernization and automation of production, the involvement of workers in the industrial environment is still high. In order to ensure workers’ well-being and a worker-friendly environment in manual assembly, it is necessary to modernize and adapt workstations to workers’ needs and implement virtual modelling approaches in order to prevent work-related injuries and diseases.

In this paper, as a solution to this problem, we propose a newly developed multi-criterial algorithm (MCA) that monitors and controls smart tools implemented on a smart manual assembly workstation. The main controlled smart tools related to the I4.0 that have an impact on increasing efficiency are a height-adjustable worktable, adjustable lighting according to direction and intensity, rotational assembly nest, and self-adjusting grab containers that adjust according to distance and tilt. The smart manual assembly workstation is also equipped with digital instructions that show the structure of the product (assembly steps) and pick-by-light technology that uses a laser beam to guide the worker to the grab containers where the required part is stored. The MCA configures the manual assembly workstation according to three groups of influential parameters: the individual worker (body height, gender), the complexity of the assembly process (precise, normal, heavy), and the type of product (dimensions, product structure).

The MCA was tested with experimental analysis in a laboratory environment, where we compared the times of manual assembly in a smart and a classic assembly workstation to show the increase in efficiency, productivity, and time savings. In the virtual environment, we performed several ergonomic analyses to show the suitability of the working conditions during the manual assembly process for the individual worker. The performed analyses were related to the reach envelope, the distribution of the parts in the grab containers and the forward reaches, as well as the load on the joints during the assembly process without (basic version) and with smart tools (improved version).

In the time analysis, we compared the times obtained during the assembly process on a smart manual assembly workstation controlled by an MCA with the times obtained during the assembly process on a classic manual assembly workstation. For all eight products for which we compared times, shorter times resulted when workers performed assembly at a smart manual assembly workstation controlled by a new algorithm. On average, times were 13.6% shorter. Despite the fact that we achieved shorter times for manual assembly at a smart manual assembly workstation, it must be emphasized that this is the result of the idea and purposeful development of a smart manual assembly workstation controlled by a new algorithm with the individual worker in mind, and not an intention to restrict or burden workers with new, even stricter time standards. By achieving shorter times without “official” time standards for manual assembly, we have shown that each worker performs the assembly process faster at a manual assembly workstation implemented with an MCA and smart tools, increasing the efficiency and productivity of the overall work process.

For ergonomic analyses, we used Siemens Jack simulation tool, which allows us to evaluate the suitability of the workstation and use “what-if” scenarios to identify health risk factors without actually exposing the workers.

The ergonomic reach envelope analysis showed that the manual assembly workstation is ergonomically suitable for each worker in terms of smart tool arrangement (distribution), according to the calculation of the MCA. The results of the reach envelope analysis can be interpreted as the maximum range in which the worker can reach for parts or tools without health risk. From the results of our case study, the arrangement of the grab containers calculated by the new algorithm is within the reach envelope, which means that the distribution of the smart tools is ergonomically suitable for the long-term work of an individual worker.

The ergonomic analysis of “reaching forward” has shown that the arrangement of self-adjusting grab containers according to the calculations of the new algorithm is appropriate on a smart manual assembly workstation, as the worker never enters the area C, which poses a health risk and the possibility of work-related diseases and injuries in the long term.

When analyzing the stress on the joints, we ran two “what-if” scenarios. The first represents the basic version of the manual assembly process at the workstation without implemented smart tools, and the second an improved version focusing on the assembly at the workstation with implemented smart tools and algorithms. The results show that with the improved version, the time of overloading individual joints is reduced or eliminated, which means that the assembly process is more ergonomically appropriate.

The results of all ergonomic analyses performed on an MCA, which controlled the manual assembly workstation show that any configuration of the workstation is ergonomically suitable for individual workers to perform manual assembly without the risk of work-related diseases and injuries.

All the results of the analyses have shown that the introduction of I4.0 principles, the digital transformation of existing workplaces, and the implementation of a multi-criterial algorithm at manual assembly workstations is recommended, both to preserve the health (ergonomic analysis) and to increase the efficiency (time analysis) of workers, leading to an increase in productivity and competitiveness of the entire company.

## Figures and Tables

**Figure 1 ijerph-19-03496-f001:**
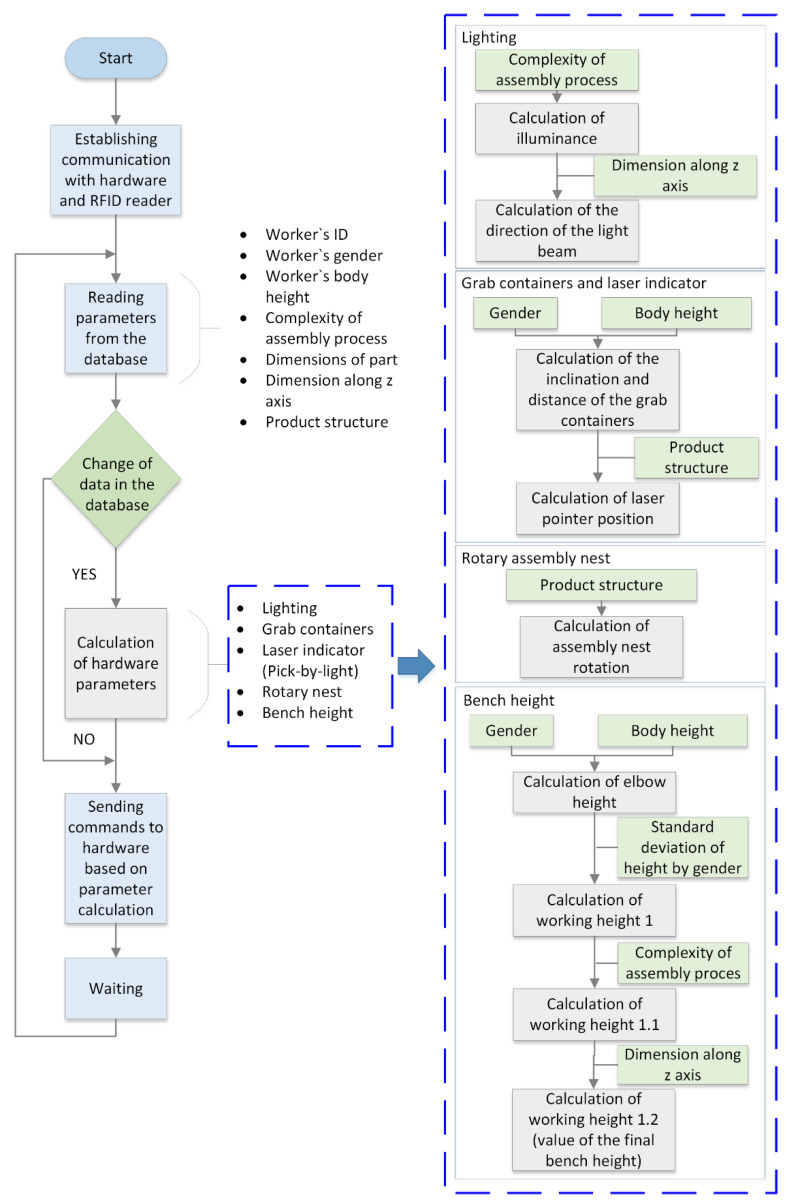
Block diagram of new multi-criterial algorithm and influential parameters.

**Figure 2 ijerph-19-03496-f002:**
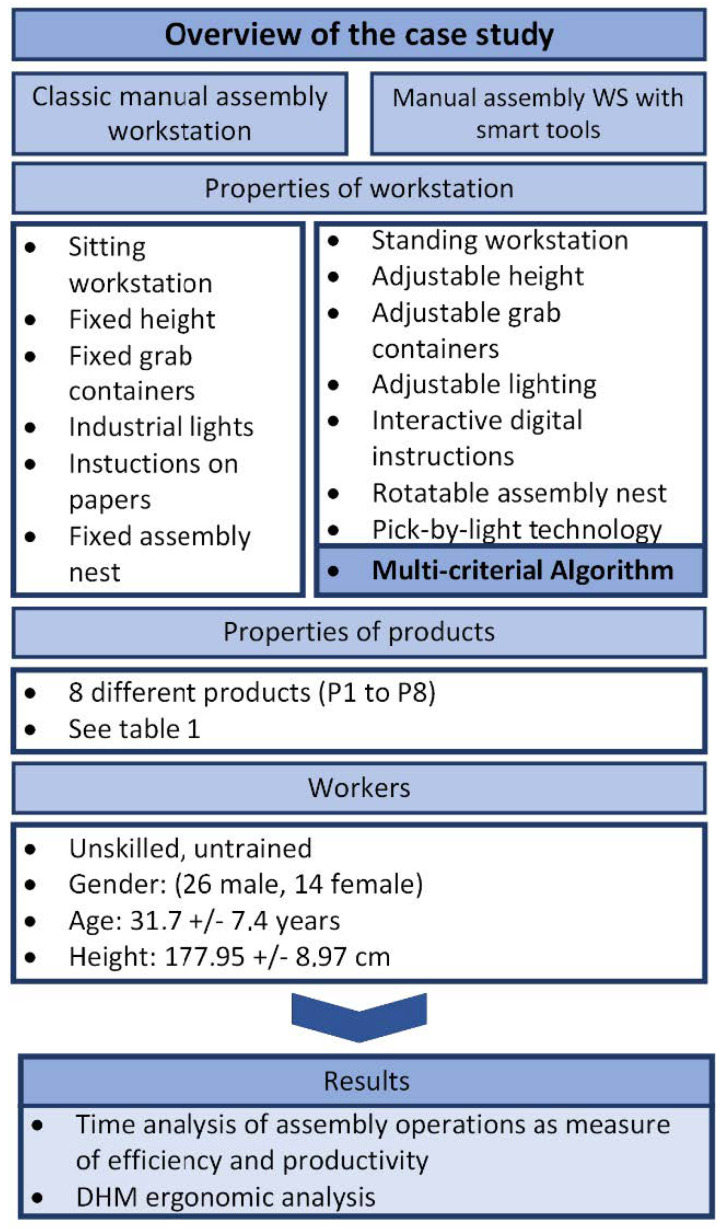
Overview of the case study.

**Figure 3 ijerph-19-03496-f003:**
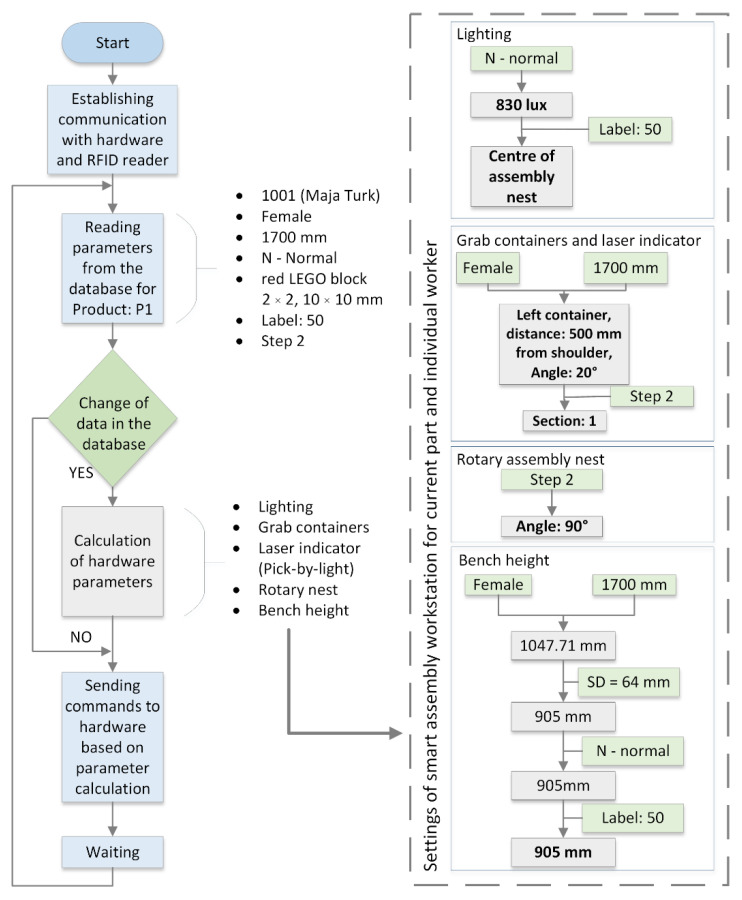
Block diagram of a multi-criterial algorithm for step 2.

**Figure 4 ijerph-19-03496-f004:**
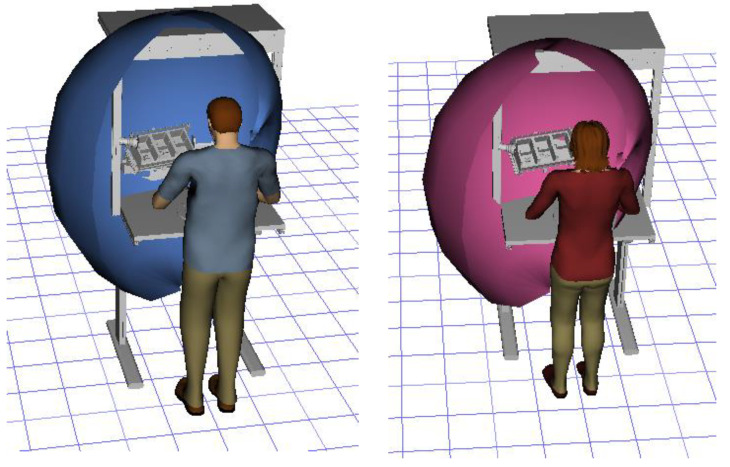
Reach analysis for male (**left**) and female (**right**) worker.

**Figure 5 ijerph-19-03496-f005:**
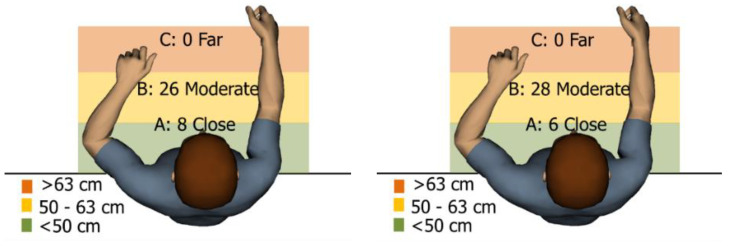
Number of forward reaches during the assembly process of product P1.

**Table 1 ijerph-19-03496-t001:** Properties of products.

Product Name	Figure of Product	Number of Parts	Number of Different Parts	Assembly Complexity	Dimension of Part	Dimension along *Z*-axis
P1	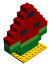	34	12	N ^1^	60 × 30 mm	50 mm
P2	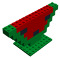	25	6	N	60 × 30 mm	50 mm
P3	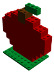	31	6	N	60 × 30 mm	50 mm
P4	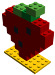	23	8	N	60 × 30 mm	50 mm
P5	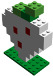	27	10	N	60 × 30 mm	50 mm
P6	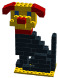	61	18	N	75 × 30 mm	150 mm
P7	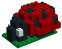	65	14	N	75 × 60 mm	50 mm
P8	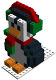	121	29	N	60 × 60 mm	150 mm

^1^ N = normal assembly.

**Table 2 ijerph-19-03496-t002:** Result of the time analysis.

Product Name	Final Time (SD) (s)		Time Saving [%]
	Classic Workstation	Smart Workstation	
P1	216.41 (41.5)	185.3 (23.5)	14.4
P2	163.7 (21.6)	142.5 (18.3)	13.0
P3	187.4 (39.2)	159.7 (31.6)	14.8
P4	154.8 (25.7)	131.8 (15.9)	14.9
P5	160.7 (36.2)	139.2 (28.4)	13.4
P6	405.1 (63.7)	345.7 (49.7)	14.7
P7	457.6 (98.1)	401.2 (56.8)	12.3
P8	1082.3 (121.4)	959.8 (87.1)	11.3

**Table 3 ijerph-19-03496-t003:** Result of the analysis of the joints’ strain during the assembly process without (basic version) and with (improved version) smart tools implemented on the manual assembly workstation.

Joint Name	Basic Version	Improved Version
Joint: Neck			
Flexion	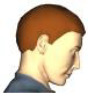	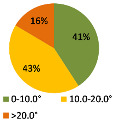	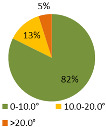
Joint: Back			
Flexion	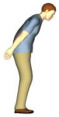	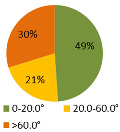	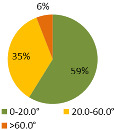
Joint: Left Shoulder			
Flexion	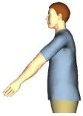	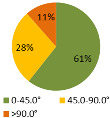	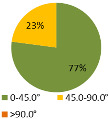
Abduction	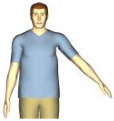	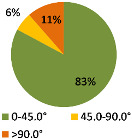	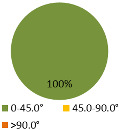

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
