# Peer review of "Multi-Criterial Algorithm for the Efficient and Ergonomic Manual Assembly Process"

_ijerph, 2022, doi:10.3390/ijerph19063496_

Round 1

Reviewer 1 Report

The paper has a very high rate of plagiarism than normal standard.

Author Response

Dear Reviewer,

Thanks very much for taking your time to review this article. I really appreciate all your comments and suggestions! Please find my itemized responses in the attachment Response to reviewers and my revisions/corrections in the re-submitted files (using “Track Changes” function).

Thanks again!

Reviewer 2 Report

The article presents the results of the implementation of a multi-criterial algorithm (MCA) for working processes and digital transformation of existing workplaces.

The results are clearly presented and methods are adequately described.

The Scientific Soundness of the material may be improved by Scientific hypothesis formulation, math formalization of the new MCA and more quantitative results of the MCA implementation

Author Response

(The authors gave the same response as above.)

Reviewer 3 Report

Interesting paper, well written and easy to follow. However some statements need to be clarified / corrected.

The introduction is well written and presents important past studies in the field. However the presentation is rather descriptive and does not have a critical element.

Definition of assembly as given in lines 34-36 is not proper.

Statement on the research gap in lines 139-141 is not supported by evidence presented before.  The necessitty for developing a new algorithm is not explained, when algorithms as such have not presented critically (their advantages, limitations etc).

if the goal is to develop an algorittm that improves the ergonomics of an assemply workplace, shouldn't there be (at least) discussion of the various frameworks used for assessing the ergonomy of a workplace and then use one to validate the effectiveness of the proposed solution?

Materials & Methods

Figure 1 is very good and clearly shows the process flow (the left side). However, there seems to be a lack of connection between the left and right side of the diagram. It might be better to split it into two diagrams and each is discussed in detail.

The case study is interesting.  However, it would be interesting if the "operators" were not volunteers but experienced ones.  The "assembly complexity" in table 1 is not explained. What is the reason to report this, if all have the same level of this metric?

Results and discussion / conclusions

again well written.  Major concern: the link to I4.0 is vere weak.  Just because I4.0 is a "trendy" term, it should not be used if the work is not really using I4.0 technologies/tools.

Minor comment:

Industry 4.0, Industrie 4.0 and I4.0:  the authors should be consistent throughout the paper and not use all three 

Author Response

(The authors gave the same response as above.)

Round 2

Reviewer 1 Report

The author has revised the requirements of the reviewers. I think the paper now is scientific enough to publish.